# Using Eye Tracking to Assess Gaze Concentration in Meditation

**DOI:** 10.3390/s19071612

**Published:** 2019-04-03

**Authors:** Kang-Ming Chang, Miao-Tien Wu Chueh

**Affiliations:** 1Department of Photonics and Communication Engineering, Asia University, Taichung 41354, Taiwan; 2Department of Medical Research, China Medical University Hospital, China Medical University, Taichung 40402, Taiwan; 3Sacred Light Heart Chan Association, Taipei 10351, Taiwan; miaotein@buddhachan.org

**Keywords:** Gaze concentration, Heart Chan Meditation, Eye tracker

## Abstract

An important component of Heart Chan Meditation is gaze concentration training. Here, we determine whether eye tracking can be used to assess gaze concentration ability. Study participants (n = 306) were requested to focus their gaze on the innermost of three concentric circles for 1 min while their eye movements were recorded. Results suggest that participants with high scores on gaze concentration accuracy and precision had lower systolic blood pressure and higher sleep quality, suggesting that eye tracking may be effective to assess and train gaze concentration within Heart Chan Meditation.

## 1. Introduction

Because of the recent advancement of electronic information media, people are being bombarded with large amounts of information daily and must process different types of information simultaneously. Whereas the information available in the past was often inadequate, the information available today is excessive, making it increasingly difficult for people to focus on any single message. This results in insufficient concentration among children at the learning stage because they are easily distracted from learning materials by other media and computer games. Consequently, focus has become a widely discussed topic. The 14th chapter of Tang Wen (The Questions of Tang) by Liezi, who was a thinker during the Warring States period in ancient China, describes the story of how Ji Chang learned archery. Ji Chang’s teacher first asked him to learn to stare at the target without blinking, after which he learned to magnify extremely small objects and see blurred objects clearly. Ji Chang could learn archery only after mastering these two skills. After 5 years, he had mastered the two basic skills and begun learning archery, after which he went on to become an expert archer. In recent years, numerous activities have been proposed for the practice of concentration, one of which that attracts particular attention is meditation. Different meditation practices have slightly different emphases, and this study focused on Heart Chan Meditation. Beginners in Heart Chan Meditation adopt a similar practice to the aforementioned archer’s concentration practice; practitioners are required to maintain their focus on a fixed point for a long time without blinking. The subsequent training requires practitioners to close their eyes and focus on their inner body and emotions, producing resonance with inner spiritual energy through concentration, thereby achieving improvement of health and emotions. Subsequently, the practitioners enter samadhi to find inner spirituality and engage in more in-depth learning. Many studies have discussed the principles of its practice as well as its positive physical and psychological effects [1]. Training in the initial stage of Heart Chan Meditation requires participants to focus their eyes on a fixed point for a long time without blinking. For example, a practitioner might draw a small dot on the wall and focus their eyes upon it. Another practice is the famous one-finger Chan training, which involves the practitioner holding a straightened index finger at an appropriate distance in front of the eyes and focusing all attention on the tip of the index finger without blinking. Some practitioners soon feel capillary blood flow within their fingertips, a regular expansion and contraction that is synchronized with their heartbeat. Alternatively, the fingertip may feel hot or as if electricity is passing through it. After becoming proficient at this practice, the practitioner should close their eyes while remaining motionless [2] and focus on their internal self, that is, focus on the body’s organs or brain and reduce thoughts to enhance the internal energy of the organ of focus, thereby improving physical health while having positive psychological effects [3].

The Heart Chan method of staring and concentration differs slightly from the “attention” that is described in the general field of psychology, which defines five categories of attention, namely focused, selective, sustained, alternating, and divided [4]. Focused attention is to focus on a stimulus. Sustained attention is to focus on activity over a long period of time. Selective attention is the ability to focus on activity with other distracting stimuli. Alternating attention is to focus attention between two or more stimuli. Divided attention is to attend different stimuli at the same time. The five types of attention are difficult to distinguish in life, with two or more types generally interacting at any given time. Therefore, numerous tools (mostly questionnaire based) have been designed to measure attention, and instrument-assisted measurement tools have subsequently been developed. For example, the Continuous Performance Test (CPT) [5], classic Stroop test [6], the Test of Variables of Attention [7], classic Conners (CPT) test [8] and the Hooper Visual Organization Task [9]. Some commercial attention software programs can also obtain satisfactory results [10]. Attention features have numerous applications and play essential roles in sleep [11], Alzheimer’s disease [12], attention deficit hyperactivity disorder [13,14], driving [15], learning, and memory [16]. Other developed measurement methods include the application of tools that use brain waves [17,18], eye blink rate [19], positron emission tomography [20], magnetic resonance imaging [21], visual event-related potential [22], and near-infrared cameras [23]. 

Eye tracking is a popular method for measuring attention and is widely recognized because measurement using eye tracking is easy, rapid, and effective [24,25,26,27,28]. Currently, eye trackers are primarily used for measuring sustained attention, together with games, or for specific groups such as attention of sports players. The concentration practice of Heart Chan Meditation cannot be classified as practicing a specific attention type. Concentrating on a fixed point is related to focused attention, whereas continuous long-term concentration is related to sustained attention. Additionally, the ability to concentrate on the area of focus while remaining unmoved by other stimuli and disturbances—an ability developed through long-term practice of Heart Chan Meditation—is related to selective attention. Heart Chan Meditation emphasizes the present and that an individual can face and process the current environment at any time, as well as change the environment and mood; these aspects are related to alternating attention. Therefore, specific measurement systems and indicators are required to assess the concentration practice of Heart Chan Meditation. Eye tracking could have clear advantages in this context. The external manifestation of a gaze is an indication of concentration, whereas internal physical and psychological condition is reflected in sleep quality and autonomic nerve activity. The relationship between concentration of gaze, sleep quality, and autonomic nerve activity has not previously been investigated in detail and is a topic worthy of discussion. Therefore, this study had two objectives: (1) to use an eye tracker to establish indicators of concentration of gaze; and (2) to investigate the correlation between sleep quality, autonomic nerve activity, and concentration of gaze. 

## 2. Materials and Methods

### 2.1. Data Recording

In this experiment, data were obtained from the employees of numerous companies and community residents. After acquiring the participants’ consent, a small briefing session was conducted to explain the physiological significance of the experimental procedure and data, after which data were collected individually in a quiet space arranged locally. Two sets of physiological signals (eye movement and autonomic nerve activity parameters) were obtained, and the participants were requested to complete the Pittsburgh Sleep Quality Index (PSQI) questionnaire. Eye tracking was conducted using the Mangold eye tracker, which including a Software Package and Eye Tracking Hardware [29]. Eye Tracking Hardware is VT3 mini Eye Tracker, with 60 Hz speed. The Tracking distance is approximate 50–70 cm. Accuracy is around 0.5 °. Tracking Method is binocular tracking. Software Package is MangoldVision, which can record gaze data, analyze data with areas of interest (AOI) and support visualization via tools such as focus maps and heat maps. Raw data is also exported for further analysis. Before eye tracker recording, data was calibrated. The participant was asked to sit approximately 50–60 cm from a computer screen and look unblinkingly at the red dot on the screen, with the dot moving between the four corners and the center of the screen. The eye tracker recorded the pupil trajectory at these calibration positions and could then calculate the pupil trajectory for any position on the screen. The participant was asked to close their eyes and rest for more than 10 s after calibration had been completed, and the formal test was then conducted after confirming that the participant’s eyes were not tired. In the formal test, the participant was requested to perform a 1-min gaze test, in which they were to focus on the inner circle of three differently colored concentric circles for 1 min. After the measurement had been performed, a program written using R language was used to extract the required parameters from the data obtained using the eye tracker. The raw eye-tracking data comprised the two-dimensional coordinates of the gaze trajectory as well as the dwell time of each gaze point. Cardiovascular physiological parameters were measured before the eye-tracking test for half the participants and after the test for the other half. Measurement of the cardiovascular physiological parameters was performed using ANSWatch and lasted 7 min, during which the participants were seated and wore the device on their left hands, with their arms at approximately the same height as their heart. The participant then completed the PSQI questionnaire, which took approximately 5–10 min. Between 5 and 20 participants were recruited from each company or community, and a total of 442 participants from 32 companies and communities were recruited. Excluding cases in which an instrument did not successfully record data, questionnaires were incomplete, and number of occurrences of arrhythmia > 10, the remaining number of valid data sets was 306, corresponding to 157 male and 149 female participants. These participants were aged 20–77 (mean = 39.17, standard deviation = 12.95) years. The experiment was approved by the Asia University Medical Research Ethics Committee (IRB No: 10505001). 

### 2.2. Eye Tracker Feature Definition

Figure 1a shows the three differently colored concentric circles used in the eye-tracking experiment. By employing concentric circles, this experiment borrowed from the concept of target shooting, in which high concentration of bullets hitting the bullseye indicates high shooting proficiency. Concentric circles were used rather than other images, such as characters and landscapes, to exclude inner psychological factors that might have interfered with the participants’ gaze. By borrowing from the concept of target shooting, this experiment also introduced the concepts of gaze precision and accuracy. Gaze accuracy is the degree to which a participant can focus on the inner circle, whereas gaze precision is the degree of which the gaze trajectories are concentrated. Four parameters were proposed on the basis of these concepts, together with the essential concentration concepts in Heart Chan Meditation. The first two parameters are related to gaze accuracy: (1) the time of the inner circle, equal to the sum of time a participant’s gaze was on the inner circle; and (2) the longest continuous inner circle viewing time, which was calculated by constructing a time series of the gaze trajectories in the inner circle, from which the longest continuous inner circle viewing time was extracted. The time resolution of the eye tracker is 0.016 s; thus, even if a participant was unconsciously distracted and shifted their gaze to outside the inner circle for only 0.016 s, their gaze was identified as interrupted. Accordingly, the longest continuous inner circle viewing time was employed as one parameter. Higher gaze accuracy resulted in a higher time of the inner circle and longer continuous inner circle viewing time. The remaining two parameters, (3) focus radius and (4) maximum saccade distance, are related to gaze precision. The focus radius was acquired by first calculating the center of mass (CM) of all gaze trajectory positions, followed by calculating the distance between each trajectory position and the CM. The acquired distances were deemed as time series data and arranged in order from least to greatest for calculation of median, thereby obtaining the focus radius. High gaze precision indicates a short focus radius. The fourth parameter was the maximum distance between the positions of adjacent gaze trajectories, known as the saccade distance. High gaze precision indicates a short saccade distance. The maximum saccade distance was used as the parameter because it indicates the greatest degree of distraction during the gaze concentration process. This parameter indicates the stability of a participant’s gaze; a more stable gaze yields a smaller maximum saccade distance. The mathematical formulae expressing the aforementioned parameter definitions are displayed in Appendix A. 

### 2.3. Cardiovascular and Sleep Quality Parameters

The cardiovascular physiological parameters of the participants were acquired using ANSWatch. The collected data were the systolic blood pressure (SYS), diastolic blood pressure (DIA), pulse pressure, number of heartbeats per minute (HR), number of occurrences of arrhythmia, Activity of the autonomic nerve (HRV), low frequency spectrum power of the autonomic nerve system (LF), high frequency spectrum power of the autonomic nerve system (HF), and biological age of the autonomic nerve activity (biological HRV age). Subsequently, the LF power ratio of the autonomic nerve system (LF%) was calculated using the LF and HF power. Sleep quality was measured using the PSQI questionnaire, a self-report questionnaire that assesses sleep quality over the previous month. It consists of 19 questions and can be completed in approximately 5 min. The total score is 21, with a lower score indicating higher sleep quality. The most commonly used cutoff point is 5, with a score of lower than 5 indicating that the participant does not have any sleeping problems. The performance of Chinese version of PSQI had been evaluated. According to Pei-Shan Tsai et.al study, the overall reliability coefficient of Chinese version of PSQI is around 0.82–0.83 for all subjects. The test–retest reliability coefficient over a 14- to 21-day interval for all subjects is 0.85. CPSQI of greater than 6 resulted in a sensitivity and specificity of 90 and 67%. Their finding showed that Chinese version of PSQI is a sensitive, reliable, and valid outcome assessment tool for use in community-based studies of primary insomnia [30].

### 2.4. Statistics

(a) Descriptive statistics

The minimum, first quartile (Q1), median, third quartile (Q3), maximum, mean, and standard deviation of each of the four eye tracker parameters were calculated. For the cardiovascular parameters and PSQI score, the mean and standard deviation only were determined. 

(b) Correlation analysis and regression

The pairwise coefficients of correlation between the four eye tracker parameters were calculated by carrplots of R. 

(c) Statistical verification

After dividing the participants into five groups on the basis of their eye tracker accuracy and precision, the statistical differences in cardiovascular and sleep parameters among the five groups were calculated by ANOVA, and differences among each group pairs were estimated by t-test. Additionally, after dividing the participants into two groups using PSQI = 5 as the cutoff point, the eye tracker parameters for the two groups were compared by t-test. The value of significance was set to 0.05. All statistics is operated by R.

## 3. Results

### 3.1. Verification of the Features Defined for Eye Tracking

Figure 2a presents the data for the four features, expressed in quartiles. The median time of the inner circle was approximately 50 s given the total recording time of 60 s. However, the median longest continuous inner circle viewing time was approximately 3.5 s, indicating that paying attention to the inner circle for a long time was difficult because the participants were easily distracted. Despite there being no positive linear relationship between the two accuracy-related parameters, a longer time of the inner circle generally corresponded with a higher longest continuous inner circle viewing time. According to Figure 2b, a greater maximum saccade distance corresponded with a greater focus radius. Regarding the precision-related parameters, a crucial question is how small the focus radius and maximum saccade distance are. For a person with high precision in gaze concentration, these values are extremely small; however, the limits of these variables are unknown. 

Figure 3 shows the pairwise correlation coefficients for the four eye tracker features. 

Regarding the accuracy-related parameters, the coefficient of correlation between time of the inner circle and longest continuous inner circle viewing time (seconds) was 0.505; regarding the precision-related parameters, the coefficient of correlation between the maximum saccade distance and focus radius was 0.526. Both correlation coefficients are high. On the other hand, coefficients of correlation between the precision and accuracy parameters were negative, which indicated moderate and weak correlations between the four parameters. 

The gaze concentration accuracy score and precision score of each participant were calculated using Equations (A9) and (A10). The scores were then plotted, as displayed in Figure 4. The accuracy and precision scores were grouped as follows to determine a participant’s gaze concentration score: Group 1, excellent: the accuracy and precision scores were both >80, accounting for approximately 8.5% of all subjects; Group 2, satisfactory: the accuracy and precision scores were both >60 (excluding scores that were >80), accounting for approximately 19.3% of all subjects; Group 3, average: the accuracy and precision scores did not belong to any other group, accounting for approximately 44.34% of all subjects;Group 4, below average: the accuracy and precision scores were both <40 (excluding scores that were both <20), accounting for approximately 19.3% of all subjects;Group 5, unsatisfactory: the accuracy and precision scores were both <20, accounting for approximately 8.5% of all subjects 

The physiological parameter differences of the different gaze concentration groups were subsequently analyzed. To determine whether the accuracy and precision scores of gaze concentration were related to the participants’ ages, the coefficients of correlation between age and the accuracy and precision scores were calculated, yielding results of 0.018 and −0.001, respectively. This indicates that the accuracy and precision scores of gaze concentration were unrelated to the participants’ ages. 

### 3.2. Verification of Cardiovascular and Sleep Quality Features

Table 1 shows the findings for the cardiovascular and sleep quality features obtained at the same time point. The obtained SYS, DIA, HR, and LF% did not exhibit any strong variation and were similar to generally reported values in healthy people. ANOVA testing and t-test between each group pairs were also examined, and there was no significance result of ANOVA testing. Further t-testing between each groups, significant differences were identified in the SYS and PSQI between group 1 (excellent) and group 5 (unsatisfactory). Group 1 had lower blood pressure and higher sleep quality than group 5.

The average PSQI was 8.5. Using PSQI = 5 as the cutoff point, 66 participants were discovered to have a PSQI score of 1–5, which was 21.5% of all the participants. The distributions of the gaze concentration parameters between the groups with PSQI ≤ 5 and PSQI > 5 were compared in Table 2. The data also indicate no significant differences between the two groups in terms of the various gaze concentration parameters. The correlations between the eye-tracking parameters, HRV, and PSQI were also investigated. Calculation of the correlation coefficients of the four eye tracker parameters with HRV and PSQI yielded coefficients that were <0.1, indicating no correlations.

## 4. Discussion 

The original goal of this study was to assist Heart Chan Meditation practitioners in establishing objective quantitative indicators for assessing gaze concentration. The participants were requested to focus on the inner circle for 60 s. However, most of the participants focused on the inner circle for a total of less than 50 s over this period; that is, the gaze of the participants would slip outside the inner circle for more than 10 s in total. The target participant action in this study was fixation. Fixation was further divided into tremors, drifts, and microsaccades [31], among which microsaccades are saccades produced when fixation is required [32]. The gaze of the participants should thus have been more commonly microsaccades. One study on fixation showed that microsaccades occur at least 2–3 times a second, and the measurement time of a microsaccade is 0.5–2.5 s [33]. The measurement time in the present study was 60 s, and the resulting data indicated large distance movements, which are the range of saccades. Additionally, 500 to 2000 Hz eyetrackers have generally been used to detect microsaccades, whereas the eyetracker used in this study had a frequency of 120 Hz and was thus unable to adequately distinguish tremors, drifts, and microsaccades. Otero-Millan et al. reported that distinguishing between saccades and microsaccades according to their amplitude or any other physical parameter is impossible [34]. Both microsaccades and saccadic intrusions during fixation should be referred to as fixation saccades [35]. Therefore, the data in this study may reflect a combination of all three fixation types. However, the parameters extracted on the basis of accuracy and precision in this experiment could encompass saccades and fixation movements, and could meet the needs of Heart Chan Meditation practitioners. Further comparative studies are required to determine the correlation of gaze concentration with saccades, microsaccades, and fixational eye movements. 

This study used an eye tracker to quantify gaze concentration and proposed four gaze concentration parameters using the concepts of accuracy and precision. The four parameters are only weakly correlated and, in combination, effectively reflect the accuracy and precision of gaze concentration. This was impossible with conventional questionnaire-based measurement tools. Because the participants in this experiment were asked to focus on the inner of three differently colored concentric circles, future researchers may have different opinions on the colors and the ratio of concentric circle radii that should be used. Whether differently shaped targets affect the findings could be investigated. One study employed different target shapes for concentration practice, and the combination of a bullseye and crosshair was discovered to result in both low dispersion and microsaccade rates [36]. Scholars also showed that blurring, color, luminance, and/or luminance contrast have no effect on fixational eye movements unless they render the target barely visible, in which case the fixation is poor [37]. The three concentric circles employed in the present study were similar to the bullseye design in the aforementioned study. The effects of the experimental method in this study on microsaccade rate cannot be evaluated. However this study proposed a new direction for defining parameters related to gaze concentration, and different colors and radius ratios can be employed to determine their effect on the four parameters. This is why the absolute parameter values were re-expressed as ordered accordingly, which should have eliminated any effect of the particular concentric circles employed on the values of the four parameters. 

Two of the parameters, namely time of the inner circle and longest continuous inner circle viewing time, were related to the precision of gaze concentration and thus were used to determine a precision score. The time of the inner circle was the first parameter can facilitate the observation of a group with below-average ranking percentage in terms of gaze concentration accuracy. The group members in the top 25% for gaze concentration accuracy were distracted from the inner circle for 3.5 s in total, resulting in difficulty differentiating between members of the group with high gaze concentration accuracy. Hence, the second accuracy parameter—longest continuous inner circle viewing time—was also employed. The time range of the longest continuous inner circle viewing time from Q3 to maximum are large, that enabling easy differentiation of those in the upper quartile for gaze concentration accuracy. Therefore, the combination of the two parameters was complementary, improving differentiation at the two extremes of high and low gaze concentration accuracy. For the similar reason, the longest continuous inner circle viewing time is skewed toward low values because a participant’s gaze could easily drift unconsciously and involuntarily to areas outside of the inner circle within 0.01 s. Therefore, this parameter is only suitable for assessing people with high gaze concentration accuracy. Use of this parameter was inspired by the method used in Heart Chan Meditation, which requires learners to focus on a fixed point without moving their gaze. This study used gaze concentration to investigate the participants’ physiological changes. The two parameters for assessing gaze concentration precision were not as extreme as the two accuracy-related parameters. The focus radius and maximum saccade distance were similar in terms of their Q1, median, and Q3; only the group in the lower quartile (i.e., bottom 25%) exhibited divergence in its precision. The differences between the accuracy- and precision-related parameters are clearly reflected in Figure 2, which indicates that some participants with large maximum saccade distances may have had low focus radii, which explains why both accuracy and precision have to be represented by at least two parameters. 

The accuracy and precision scores of gaze concentration proposed in this study can quantify gaze concentration, which is unrelated to age. This is interesting because it means that older people do not necessarily have poorer ability to focus. This study also proposed a standard of gaze concentration grouping. On the basis of the score distributions of gaze concentration accuracy and precision, five groups can be defined—excellent, satisfactory, average, below average, and unsatisfactory respectively, in this study. Analysis of the physiological parameters of these groups revealed that Group 1 (excellent concentration group) had a significantly lower PSQI (higher sleep quality) and a lower SYS than did Group 5 (unsatisfactory concentration group). Although the evidence for this result is insufficient and more research is required for verification, ancient Chinese texts and traditional Chinese medicine note that a person with a bright and spirited gaze is likely to be in excellent physical health, which would explain the high sleep quality among the high concentration group. Additionally, stable qi naturally leads to stable emotions, which in turn results in lower blood pressure. Further research should be conducted to verify these arguments. 

This study investigated concentration during Heart Chan Meditation. Eye movement has also been previously employed in studies to evaluate meditation attention. Kumari et al. used infrared oculography to examine mindfulness meditation, particularly performance during smooth pursuit eye movements, antisaccade tasks, and prosaccade tasks [38]. They measured eye closure, obtained electroencephalograms during focused breathing, and extracted eye-movement-derived ocular data from electroencephalogram signals instead of using a standard multielectrode electrooculography montage to monitor ocular data. The study by Matiz et al. discovered that when the eye is closed, the difference between mind-wandering and focus can be detected using the spectrum of eye movement [39], and this is also a satisfactory approach for detecting the degree of mind-wandering. In the present study, eyetracking parameters were used to detect the training results of focusing with open eyes. Whether the minds of the highly focused group wandered less is another research topic. Additionally, this study [39] used a combination of several focus methods: (1) mindfulness meditation, in which the practitioner focuses their attention on breathing sensations; (2) Vipassana meditation, in which the practitioner moves their attention throughout their body, scanning each part of it; (3) Isha Shoonya meditation, in which the practitioner pays attention to their thought processes in an attempt to consciously experience spontaneous trains of thought, emotions, and sensations; and (4) Himalayan Yoga, in which the practitioner mentally repeats a mantra with or without breath awareness. Another study investigated the gaze effect within Tai Chi [40]; however, this practice is not entirely the same as Heart Chan–based attention meditation. Therefore, whether the research method employed in the present study is applicable to all meditation patterns remains to be determined. 

A limitation of this study was the exclusion of the population with visual impairments such as cataracts, astigmatism, or severe myopia; people with these impairments may not be able to focus, and thus their eyes cannot be tracked using an eye tracker. Moreover, the correlations between gaze concentration ability and other behavioral observation indicators require subsequent research. However, the advantage of this study was the rapid speed of measurements because no questionnaires or behavioral observation were required to assess gaze concentration. A follow-up study can be conducted to explore the correlation of other physiological parameters and behavioral indicators in Group 1 (excellent concentration group) and Group 5 (unsatisfactory concentration group). If a significant correlation can be determined, it can be used as a tool for rapid screening. For example, human resource units in companies can screen for applicants with high concentration and assign them jobs that require a highly stable gaze. The screening can also be used to identify individuals suitable for special occupations such as being an athlete or pilot, as well as in research related to the education of schoolchildren, internet addiction, and drug abuse. 

## 5. Conclusions

This study proposed a measurement system that uses an eye tracker to measure gaze concentration ability; this ability was quantified into a gaze concentration score using concepts of gaze concentration accuracy and precision, ultimately defining five groups of ability levels. The group with excellent gaze concentration exhibited satisfactory sleep quality and blood pressure compared with the group with unsatisfactory gaze concentration. Making measurements using the proposed system is straightforward and rapid, and the method is suitable for assessing the performance of an individual at Heart Chan Meditation. Additionally, it can be used for subsequent studies on other extended topics involving gaze concentration. 

## Ethical Statements

All subjects gave their informed consent for inclusion before they participated in the study. The study was conducted in accordance with the Declaration of Helsinki, and the protocol was approved by the Asia University Medical Research Ethics Committee ((IRB No: 10505001).

## Figures and Tables

**Figure 1 sensors-19-01612-f001:**
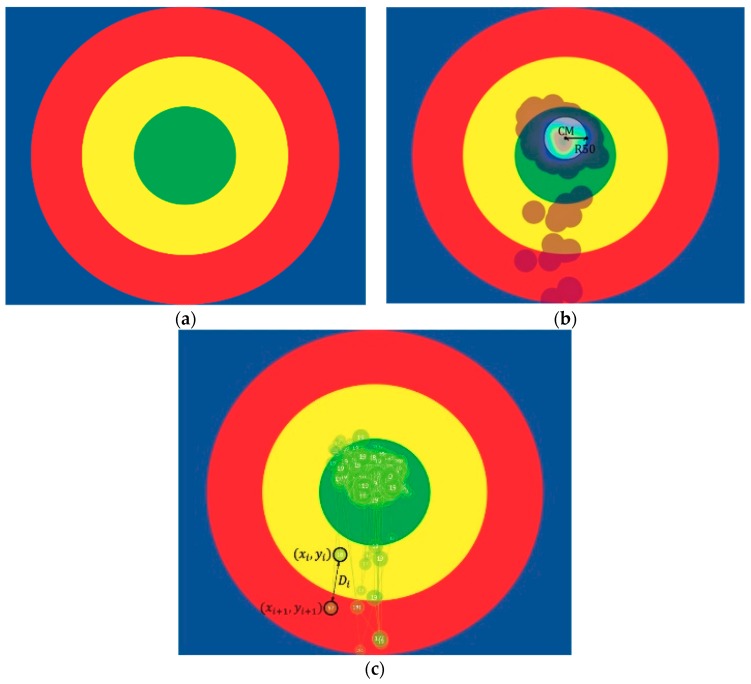
Eye tracker feature definition. (**a**) Displayed image, consisting of three concentric circles; The participant was requested to focus their gaze on the inner circle. (**b**) Eye tracker hot zones: the center is CM, and the focus radius is R50. (**c**) Eye tracker trajectory; the numbers in the small circles are the $t_{i}$. The diagram shows the coordinates at two time points and the saccade distance *Di*.

**Figure 2 sensors-19-01612-f002:**
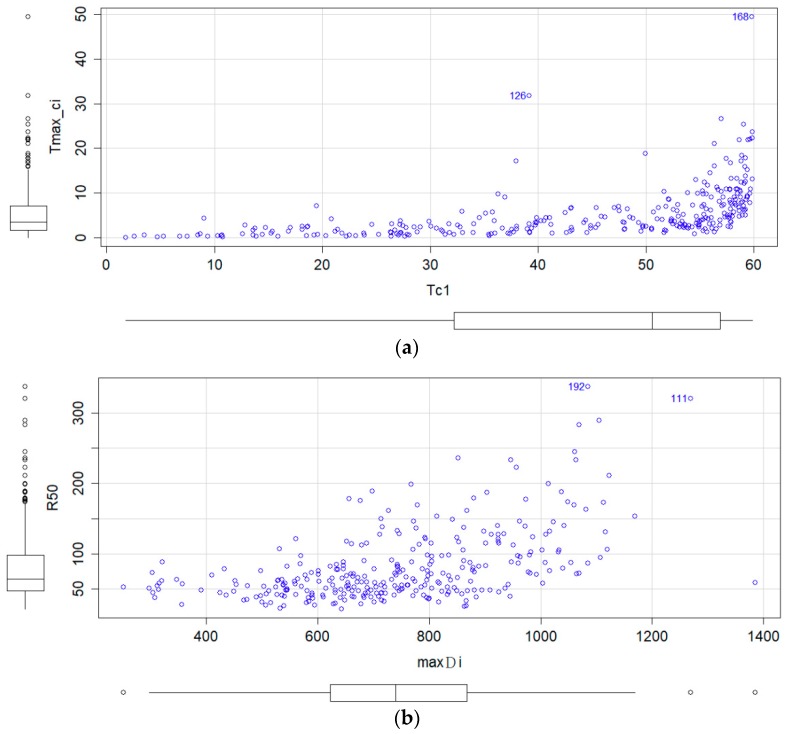
Boxplot distributions for the eye tracker features. (**a**)$\;T_{c1}$ and $T_{\max\_ci}$; (**b**) maxDi and R50.

**Figure 3 sensors-19-01612-f003:**
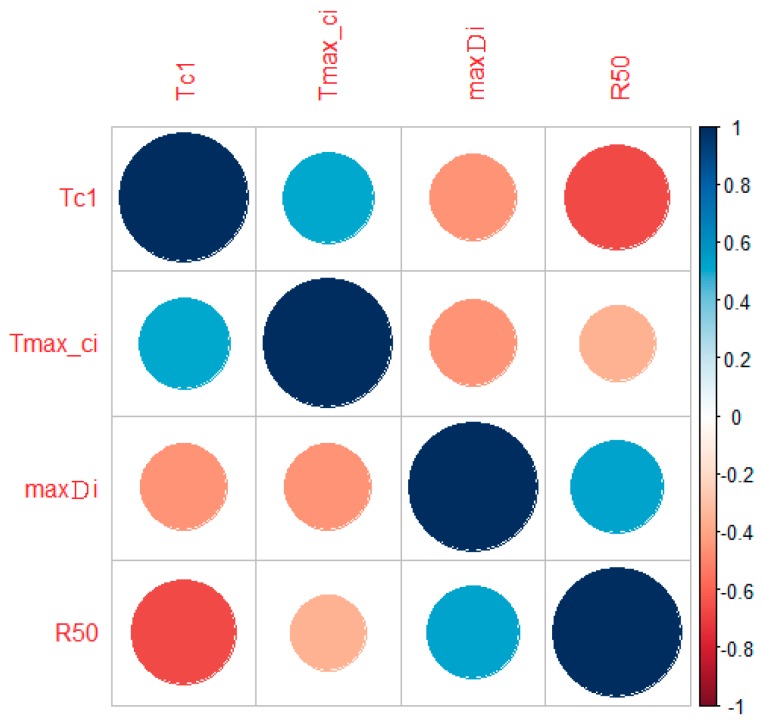
Correlation plot between the four features.

**Figure 4 sensors-19-01612-f004:**
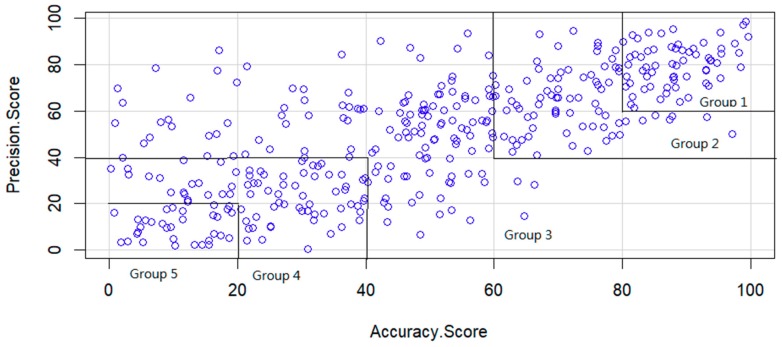
Gaze concentration precision score versus accuracy score.

**Table 1 sensors-19-01612-t001:** Cardiovascular and sleep quality feature distributions for all subjects and five eye-tracker-index-derived groups. a and b are p-value of t-testing between Group 1 and Group 5.

	All	Group 1	Group 2	Group 3	Group 4	Group 5
SYS	116.72 (15.72)	115.3 (11.3) ^a*^	114.1 (10.4)	117.3 (17.0)	115.3 (15.4)	123.4 (18.9)
DIA	80.47 (10.16)	81.8 (10.7)	79.8 (9.9)	80.1 (9.5)	79.8 (10.8)	84.1 (11.2)
HR	76.34 (11.13)	75.8 (11.9)	77.2 (10.1)	75.8 (10.6)	77.9 (13.9)	72.3 (9.6)
HRV	40.43 (17.27)	39.7 (15.3)	42.0 (17.4)	40.0 (16.6)	38.7 (17.7)	45.5 (21.5)
LF%	56.71 (19.63)	59.6 (21.6)	56.7 (21.0)	55.3 (18.7)	57.5 (18.0)	59.7 (24.2)
HF%	43.29 (19.63)	40.4 (21.6)	43.3 (21.0)	44.8 (18.7)	42.5 (18.0)	40.3 (24.2)
LF	579.13 (824.55)	551.8 (802.9)	730.6 (1230.1)	503.6 (606.2)	528.2 (702.5)	820.3 (949.0)
HF	357.79 (414.96)	279.7 (195.3)	361.7 (265.8)	369.3 (461.4)	325.0 (408.0)	465.0 (602.7)
LF/HF(%)	2.24 (3.05)	3.1 (3.9)	2.4 (3.2)	1.9 (2.1)	2.3 (4.1)	3.0 (3.2)
HRV age	38.37 (18.88)	38.3 (21.0)	36.1 (16.3)	38.7 (19.1)	40.5 (19.7)	35.0 (19.7)
Real Age	**39.17** **(12.95)**	38.0 (11.4)	39.1 (11.5)	39.0 (12.8)	40.8 (14.4)	36.3 (14.8)
PSQI	8.53 (3.65)	7.9 (3.1) ^b*^	8.7 (3.6)	8.3 (3.7)	8.9 (3.9)	9.5 (3.4)

**Table 2 sensors-19-01612-t002:** Gaze parameters between low and high PSQI groups.

Parameter	PSQI 1–5 (N = 66)	PSQI > 5 (N = 240)
$T_{c1}$	43.74 (0.15)	43.69 (0.15)
$T_{\max\_ci\_}$	4.78 (4.51)	5.51(6.12)
maxDi	740.75 (175.18)	742.81 (198.88)
R50	77.45 (41.31)	82.95 (53.40)
Accuracy score	48.64 (26.27)	50.28 (27.27)
Precision score	50.49 (24.41)	49.93 (25.3)

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
