# Peer review of "Using Eye Tracking to Assess Gaze Concentration in Meditation"

_sensors, 2019, doi:10.3390/s19071612_

Round 1
Reviewer 1 Report
There are few things that should be addressed. In first instance a summary of the result should be added to the abstract so a reader has a clear view and can quantify how much the concentration (measured as gaze precision) influence the physiology.
Secondly I would like to see (at least in the discussion section) a normalization of the sleep quality based upon the sensitivity of the self assessment test used. these could help to put in perspective the "quality of sleep assertion"
Lastly would be easier to read the paper if some of the statistical analysis table are also presented as pictorial representation i.e. box plot
Author Response
1.There are few things that should be addressed. In first instance a summary of the result should be added to the abstract so a reader has a clear view and can quantify how much the concentration (measured as gaze precision) influence the physiology.
Reply: a revised abstract is added as following “Results suggest that participants with high scores on gaze concentration accuracy and precision had lower systolic blood pressure and higher sleep quality, suggesting that eye tracking may be effective to assess and train gaze concentration within Heart Chan Meditation”
2. Secondly I would like to see (at least in the discussion section) a normalization of the sleep quality based upon the sensitivity of the self assessment test used. these could help to put in perspective the "quality of sleep assertion"
Reply: in section 2.3, following is added:
The performance of Chinese version of PSQI had been evaluated. According to Pei-Shan Tsai et.al study, the overall reliability coefficient of Chinese version of PSQI is around 0.82 –0.83 for all subjects. The test–retest reliability coefficient over a 14- to 21-day interval for all subjects is 0.85. CPSQI of greater than 6 resulted in a sensitivity and specificity of 90 and 67%. Their finding showed that Chinese version of PSQI is a sensitive, reliable, and valid outcome assessment tool for use in community-based studies of primary insomnia [30].
Reply: Eye tracker features are combined and shown in figure 2,
Reviewer 2 Report
@page { margin: 2cm } p { margin-bottom: 0.25cm; line-height: 120% }In their paper entitled "Gaze Concentration Measurement System and Quantitative Indicators for Assessing Heart Chan Meditation Performance by Using an Eye Tracker and Correlations between Physiological Parameters", the authors examine whether eye tracking can be used to determine how well participants can focus their gaze as part of a meditation technique known as Heart Chan Meditation.
Reading the title and the abstract, I was a bit worried about the quality of the paper, but reading on, I got more and more convinced that after revision, the paper may become suitable for publication in Sensors.
My main concerns are as follows:
(1) I am not sure how accurately the eye movements can be measured for the present purpose. There may be contamination from noise in the eye movement recordings, and fixational eye movements may play a role (e.g., work by Engbert & Kliegl, 2003, Vision Research). I think more information needs to be given about whether my worries about these two factors are justified.
(2) I found the results section difficult to read. There are many plots and tables. Consider using plots instead of tables (visual representation is easier to read), and combine plots in Figures (e.g., Figure 1A, 1B, etc).
(3) The results are not placed in the context of previous findings. Please find at the bottom suggestions on how to improve this.
I also had the following specific comments:
I think the title is too long. How about something like "Using eye tracking to assess gaze concentration in meditation". While this does not specify the type of meditation and the comparison with physiological parameters, it is a title that is much easier to grasp for a possible reader.
I also think the abstract is also long and difficult to read. I would recommend something shorter, along the lines of:
"An important component of Heart Chan Meditation is gaze concentration training. Here, we determine whether eye tracking can be used to to assess gaze concentration ability. 306 participants were requested to focus their gaze on the innermost of three concentric circles for 1 minute while their eye movements were recorded. Results suggest that participants with high scores on gaze concentration accuracy and precision had lower systolic blood pressure and higher sleep quality, suggesting that eye tracking may be effective to assess and train gaze concentration within Heart Chan Meditation."
Reading the introduction, a question that came to mind quickly was whether the study would be specific to Heart Chan Meditation. I had not heard of this type of meditation before, and probably many readers would not. Can you, early on, indicate why you focus on this specific form of meditation, and whether you think your results would also apply to other forms of meditation. So, first indicate that meditation is used to focus, and then indicate the specific type of meditation that you will study and why.
The formatting of the list in lines 62 in 67 is a bit unusual. Consider using a table instead, or describe the different types of attention in your own words in a normal paragraph.
I like the lines 72 to 81, where you provide a good overview of related literature without spending a lot of words on each paper.
Consider using shorter paragraphs, to make it easier for the reader to understand the text. For example, consider starting a new paragraph at line 79, with "eye tracking".
I am not familiar with the eye tracker used (Mangold Vision eye tracker). Therefore I think it is important to provide more details about the accuracy and precision of this eye tracker. Is a chin rest normally used? Does it use infra-red images of the eyes for eye tracking? Corneal reflection? What is the temporal resolution? (I just noticed this latter information was later in the text, but best introduce it together with the make of the eye tracker).
For reproducible science, it would be good to share the R script used for analysis on a website, such as Open Science Framework, Google Drive, or Github. At this point (line 116), provide more details about how the relevant eye tracking features were extracted.
To explain the measures in lines 142-154, it would be good to provide a few examples of scanpaths (measurements connected by lines) and the values of the various measures for those scanpaths, so that the reader can more easily understand what the different measures indicate.
In lines 156-157 you talk about saccades, but it is unlikely that during focusing attention, participants make large saccades. Participants may make microsaccades, but these are more difficult to detect (look at the paper by Engbert & Kliegl, 2003, Vision Research for a method).
The section in lines 162-187 I find very difficult to read. Maybe it would be better to move this information to the appendix. Most readers will be satisfied with reading the verbal description and see examples, and readers that try to replicate your work will want to see the equations. Also here sharing the R script / code for analysis would help.
In the section 'statistical verification' (line 212) you first divide your group into 5 and then into 2. It is not entirely clear to me why you first use five and then two groups.
Instead of table 1, consider using boxplots (e.g., using R), which will contain the same information, but are easier to read.
For the correlations in Table 2, consider using corrplots from R (e.g., http://www.sthda.com/english/wiki/visualize-correlation-matrix-using-correlogram). I would also recommend not repeating the information in the table in the text (only numbers in the table, refer to table from the text and talk about small or large, or significant and non-significant correlations).
For Figures 2 and 3 it would help to add illustrations to the side to indicate what the various measures indicate. Note that these plots also suggest that for some of the data Pearson correlations may not be the best to examine the association (Figure 2), and that Spearman correlations may be better (did you mention somewhere what correlation you used in Table 2?).
Line 266: Best place the regression equation alongside the best fitting line in the plot, rather than in the text.
Lines 268 - 278: Here you talk about the five groups, while previously you spoke about 2 groups. Instead of listing the numbers for each of the five groups in the text, best use a data-plot. If you are afraid to have too many data-plots, consider combining them using sub-plots (Figure 1A, 1B, etc).
Line 284: Please remind the reader what HRV and PSQI mean.
Tables 3, 4, 5, and 6. Please show these results in the form of a (bar) graph.
Line 314. Here you indicate that the four gaze parameters are only weakly correlated, but reflect gaze concentration. This sentence made me wonder whether you could use the four gaze parameters to predict such concentration in a multiple regression model, which will also provide a measure of how well concentration can be predicted overall.
Line 317. Here it may be good to refer to: https://www.ncbi.nlm.nih.gov/pubmed/23099046 who investigated the best shape of a fixation point.
Avoid repeating results in the discussion (i.e., do not include numbers). Only describe results in more general terms.
In the discussion, more strongly link your own results to past findings. Did anyone study gaze concentration in meditation before? Did they find the same? If the results are different, why do you think they may be? How is your study better or worse than other studies? Did you use better or worse measures? Some papers that I found in Google Scholar, are:
McGibbon, C. A., Krebs, D. E., Wolf, S. L., Wayne, P. M., Scarborough, D. M., & Parker, S. W. (2004). Tai Chi and vestibular rehabilitation effects on gaze and whole-body stability. Journal of Vestibular Research, 14(6), 467-478.
Kumari, V., Antonova, E., Wright, B., Hamid, A., Hernandez, E. M., Schmechtig, A., & Ettinger, U. (2017). The mindful eye: Smooth pursuit and saccadic eye movements in meditators and non-meditators. Consciousness and cognition, 48, 66-75.
Matiz, A., Crescentini, C., Fabbro, A., Budai, R., Bergamasco, M., & Fabbro, F. (2019). Spontaneous eye movements during focused-attention mindfulness meditation. PloS one, 14(1), e0210862.
Author Response
Reviewer 2
1.I am not sure how accurately the eye movements can be measured for the present purpose. There may be contamination from noise in the eye movement recordings, and fixational eye movements may play a role (e.g., work by Engbert & Kliegl, 2003, Vision Research). I think more information needs to be given about whether my worries about these two factors are justified.
Reply: in Line 275 to line 293, following are added.
The original goal of this study was to assist Heart Chan Meditation practitioners in establishing objective quantitative indicators for assessing gaze concentration. The participants were requested to focus on the inner circle for 60 seconds. However, most of the participants focused on the inner circle for a total of less than 50 seconds over this period; that is, the gaze of the participants would slip outside the inner circle for more than 10 seconds in total. The target participant action in this study was fixation. Fixation was further divided into tremors, drifts, and microsaccades [31], among which microsaccades are saccades produced when fixation is required [32]. The gaze of the participants should thus have been more commonly microsaccades. One study on fixation showed that microsaccades occur at least 2–3 times a second, and the measurement time of a microsaccade is 0.5–2.5 seconds [33]. The measurement time in the present study was 60 seconds, and the resulting data indicated large distance movements, which are the range of saccades. Additionally, 250- and 500-Hz eyetrackers have generally been used to detect microsaccades, whereas the eyetracker used in this study had a frequency of 120 Hz and was thus unable to adequately distinguish tremors, drifts, and microsaccades. Otero-Millan et al. reported that distinguishing between saccades and microsaccades according to their amplitude or any other physical parameter is impossible [34]. Both microsaccades and saccadic intrusions during fixation should be referred to as fixation saccades [35]. Therefore, the data in this study may reflect a combination of all three fixation types. However, the parameters extracted on the basis of accuracy and precision in this experiment could encompass saccades and fixation movements, and could meet the needs of Heart Chan Meditation practitioners. Further comparative studies are required to determine the correlation of gaze concentration with saccades, microsaccades, and fixational eye movements.
(2) I found the results section difficult to read. There are many plots and tables. Consider using plots instead of tables (visual representation is easier to read), and combine plots in Figures (e.g., Figure 1A, 1B, etc).
Reply: The original six tables and six figures are reduced as four tables and two figures.
(3) The results are not placed in the context of previous findings. Please find at the bottom suggestions on how to improve this.
Reply: Three references are cited in [38], [39] and [40].
4. I think the title is too long. How about something like "Using eye tracking to assess gaze concentration in meditation". While this does not specify the type of meditation and the comparison with physiological parameters, it is a title that is much easier to grasp for a possible reader.
Reply: Title changed as suggestion.
5. I also think the abstract is also long and difficult to read. I would recommend something shorter, along the lines of:
"An important component of Heart Chan Meditation is gaze concentration training. Here, we determine whether eye tracking can be used to to assess gaze concentration ability. 306 participants were requested to focus their gaze on the innermost of three concentric circles for 1 minute while their eye movements were recorded. Results suggest that participants with high scores on gaze concentration accuracy and precision had lower systolic blood pressure and higher sleep quality, suggesting that eye tracking may be effective to assess and train gaze concentration within Heart Chan Meditation."
Reply: Abstract changed as suggestion.
6. Reading the introduction, a question that came to mind quickly was whether the study would be specific to Heart Chan Meditation. I had not heard of this type of meditation before, and probably many readers would not. Can you, early on, indicate why you focus on this specific form of meditation, and whether you think your results would also apply to other forms of meditation. So, first indicate that meditation is used to focus, and then indicate the specific type of meditation that you will study and why.
Reply: Following content added in line 34-42 to explain Heart Chan meditation:
“In recent years, numerous activities have been proposed for the practice of concentration, one of which that attracts particular attention is meditation. Different meditation practices have slightly different emphases, and this study focused on Heart Chan Meditation. Beginners in Heart Chan Meditation adopt a similar practice to the aforementioned archer’s concentration practice; practitioners are required to maintain their focus on a fixed point for a long time without blinking. The subsequent training requires practitioners to close their eyes and focus on their inner body and emotions, producing resonance with inner spiritual energy through concentration, thereby achieving improvement of health and emotions. Subsequently, the practitioners enter samadhi to find inner spirituality and engage in more in-depth learning.”
7. The formatting of the list in lines 62 in 67 is a bit unusual. Consider using a table instead, or describe the different types of attention in your own words in a normal paragraph.
Reply: Rewritten as following content and shown in line 57-60.
“ Focused attention is to focus on a stimulus. Sustained attention is to focus on activity over a long period of time. Selective attention is the ability to focus on activity with other distracting stimuli. Alternating attention is to focus attention between two or more stimuli. Divided attention is to attend different stimuli at the same time.”
8.I like the lines 72 to 81, where you provide a good overview of related literature without spending a lot of words on each paper.
Reply: Thanks for appreciate.
9. Consider using shorter paragraphs, to make it easier for the reader to understand the text. For example, consider starting a new paragraph at line 79, with "eye tracking".
Reply: Corrected as suggestion
。
10. I am not familiar with the eye tracker used (Mangold Vision eye tracker). Therefore I think it is important to provide more details about the accuracy and precision of this eye tracker. Is a chin rest normally used? Does it use infra-red images of the eyes for eye tracking? Corneal reflection? What is the temporal resolution? (I just noticed this latter information was later in the text, but best introduce it together with the make of the eye tracker).
Reply: Following content added in line 99-105 to explain eyetracker:
“Eye tracking was conducted using the Mangold eye tracker, which including a Software Package and Eye Tracking Hardware[29]. Eye Tracking Hardware is VT3 mini Eye Tracker, with 60 Hz speed. The Tracking distance is approximate 50 - 70 cm. Accuracy is around 0.5 °. Tracking Method is binocular tracking. Software Package is MangoldVision, which can recorde gaze data, analyze data with areas of interest (AOI) and support visualization, such as focus map and heat map. Raw data is also exported for further analysis. Before eye tracker recording, data was calibrated.”
11. For reproducible science, it would be good to share the R script used for analysis on a website, such as Open Science Framework, Google Drive, or Github. At this point (line 116), provide more details about how the relevant eye tracking features were extracted.
Reply: The eye tracking features are listed in Appendix, line 396-421.
12. To explain the measures in lines 142-154, it would be good to provide a few examples of scanpaths (measurements connected by lines) and the values of the various measures for those scan paths, so that the reader can more easily understand what the different measures indicate.
Reply: Eye tracker feature definition is already shown in Figure-1
13. In lines 156-157 you talk about saccades, but it is unlikely that during focusing attention, participants make large saccades. Participants may make microsaccades, but these are more difficult to detect (look at the paper by Engbert & Kliegl, 2003, Vision Research for a method).
Reply: It’s a very good question. Line 275-286 are added to discuss this issue, as following:
“The original goal of this study was to assist Heart Chan Meditation practitioners in establishing objective quantitative indicators for assessing gaze concentration. The participants were requested to focus on the inner circle for 60 seconds. However, most of the participants focused on the inner circle for a total of less than 50 seconds over this period; that is, the gaze of the participants would slip outside the inner circle for more than 10 seconds in total. The target participant action in this study was fixation. Fixation was further divided into tremors, drifts, and microsaccades [31], among which microsaccades are saccades produced when fixation is required [32]. The gaze of the participants should thus have been more commonly microsaccades. One study on fixation showed that microsaccades occur at least 2–3 times a second, and the measurement time of a microsaccade is 0.5–2.5 seconds [33]. The measurement time in the present study was 60 seconds, and the resulting data indicated large distance movements, which are the range of saccades. Additionally, 250- and 500-Hz eyetrackers have generally been used to detect microsaccades, whereas the eyetracker used in this study had a frequency of 120 Hz and was thus unable to adequately distinguish tremors, drifts, and microsaccades.”
14. The section in lines 162-187 I find very difficult to read. Maybe it would be better to move this information to the appendix. Most readers will be satisfied with reading the verbal description and see examples, and readers that try to replicate your work will want to see the equations. Also here sharing the R script / code for analysis would help.
Reply: as suggested to list into Appendix-1
15. In the section 'statistical verification' (line 212) you first divide your group into 5 and then into 2. It is not entirely clear to me why you first use five and then two groups.
Reply. Corrected as 5 groups in line 193.
16. Instead of table 1, consider using boxplots (e.g., using R), which will contain the same information, but are easier to read.
Reply: Original Table-1 plus Fig2 plus Fig3 are combined as new Figure-2。
17. For the correlations in Table 2, consider using corrplots from R (e.g., http://www.sthda.com/english/wiki/visualize-correlation-matrix-using-correlogram). I would also recommend not repeating the information in the table in the text (only numbers in the table, refer to table from the text and talk about small or large, or significant and non-significant correlations).
Reply: as suggested as shown in Figure-2。
18. For Figures 2 and 3 it would help to add illustrations to the side to indicate what the various measures indicate. Note that these plots also suggest that for some of the data Pearson correlations may not be the best to examine the association (Figure 2), and that Spearman correlations may be better (did you mention somewhere what correlation you used in Table 2?).
Reply: Original Table-1 plus Fig2 plus Fig3 are combined as new Figure-2。
19. Line 266: Best place the regression equation alongside the best fitting line in the plot, rather than in the text.
Reply: Regression equation is deleted in revised form. It has no an benefit for further analysis.
20. Lines 268 - 278: Here you talk about the five groups, while previously you spoke about 2 groups. Instead of listing the numbers for each of the five groups in the text, best use a data-plot. If you are afraid to have too many data-plots, consider combining them using sub-plots (Figure 1A, 1B, etc).
Reply. Table 3,4,5,6 in previous version are combined as two new tables.
21. Line 284: Please remind the reader what HRV and PSQI mean.
Reply. Title of Section 3.2 is corrected as “ Verification of cardiovascular and sleep quality features “
22. Tables 3, 4, 5, and 6. Please show these results in the form of a (bar) graph.
Reply. Table 3,4,5,6 in previous version are combined as two new tables.
23. Line 314. Here you indicate that the four gaze parameters are only weakly correlated, but reflect gaze concentration. This sentence made me wonder whether you could use the four gaze parameters to predict such concentration in a multiple regression model, which will also provide a measure of how well concentration can be predicted overall.
Reply: Thanks for valuable suggestion. In this manuscript, there is no other reference concentration index, such as surveys, therefore there is no output measurement to build multiple regression model. This study try to establish a novel gaze concentration index.
24. Line 317. Here it may be good to refer to: https://www.ncbi.nlm.nih.gov/pubmed/23099046 who investigated the best shape of a fixation point.
Reply: Following are added in Line 299-309 to discuss shape of fixation”
“Whether differently shaped targets affect the findings could be investigated. One study employed different target shapes for concentration practice, and the combination of a bullseye and crosshair was discovered to result in both low dispersion and microsaccade rates [36]. Scholars also showed that blurring, color, luminance, and/or luminance contrast have no effect on fixational eye movements unless they render the target barely visible, in which case the fixation is poor [37]. The three concentric circles employed in the present study were similar to the bullseye design in the aforementioned study. The effects of the experimental method in this study on microsaccade rate cannot be evaluated. However this study proposed a new direction for defining parameters related to gaze concentration, and different colors and radius ratios can be employed to determine their effect on the four parameters. This is why the absolute parameter values were re-expressed as ordered accordingly, which should have eliminated any effect of the particular concentric circles employed on the values of the four parameters. “
25. Avoid repeating results in the discussion (i.e., do not include numbers). Only describe results in more general terms.
Reply: discussion is rewritten as suggestion.
26. In the discussion, more strongly link your own results to past findings. Did anyone study gaze concentration in meditation before? Did they find the same? If the results are different, why do you think they may be? How is your study better or worse than other studies? Did you use better or worse measures? Some papers that I found in Google Scholar, are:
McGibbon, C. A., Krebs, D. E., Wolf, S. L., Wayne, P. M., Scarborough, D. M., & Parker, S. W. (2004). Tai Chi and vestibular rehabilitation effects on gaze and whole-body stability. Journal of Vestibular Research, 14(6), 467-478.
Kumari, V., Antonova, E., Wright, B., Hamid, A., Hernandez, E. M., Schmechtig, A., & Ettinger, U. (2017). The mindful eye: Smooth pursuit and saccadic eye movements in meditators and non-meditators. Consciousness and cognition, 48, 66-75.
Matiz, A., Crescentini, C., Fabbro, A., Budai, R., Bergamasco, M., & Fabbro, F. (2019). Spontaneous eye movements during focused-attention mindfulness meditation. PloS one, 14(1), e0210862.
Reply: Following is added line 348-368
“This study investigated concentration during Heart Chan Meditation. Eye movement has also been previously employed in studies to evaluate meditation attention. Kumari et al. used infrared oculography to examine mindfulness meditation, particularly performance during smooth pursuit eye movements, antisaccade tasks, and prosaccade tasks [38]. They measured eye closure, obtained electroencephalograms during focused breathing, and extracted eye-movement-derived ocular data from electroencephalogram signals instead of using a standard multielectrode electrooculography montage to monitor ocular data. The study by Matiz et al. discovered that when the eye is closed, the difference between mind-wandering and focus can be detected using the spectrum of eye movement [39], and this is also a satisfactory approach for detecting the degree of mind-wandering. In the present study, eyetracking parameters were used to detect the training results of focusing with open eyes. Whether the minds of the highly focused group wandered less is another research topic. Additionally, this study [39] used a combination of several focus methods: (1) mindfulness meditation, in which the practitioner focuses their attention on breathing sensations; (2) Vipassana meditation, in which the practitioner moves their attention throughout their body, scanning each part of it; (3) Isha Shoonya meditation, in which the practitioner pays attention to their thought processes in an attempt to consciously experience spontaneous trains of thought, emotions, and sensations; and (4) Himalayan Yoga, in which the practitioner mentally repeats a mantra with or without breath awareness. Another study investigated the gaze effect within Tai Chi [40]; however, this practice is not entirely the same as Heart Chan–based attention meditation. Therefore, whether the research method employed in the present study is applicable to all meditation patterns remains to be determined.”
Round 2
Reviewer 1 Report
I am satisfied by the author responses and recommend publication